# BREAst screening Tailored for HEr (BREATHE)—A study protocol on personalised risk-based breast cancer screening programme

Jenny Liu[1,2], Peh Joo Ho[2,3,4], Tricia Hui Ling Tan[3], Yen Shing Yeoh[3], Ying Jia Chew[1,5], Nur Khaliesah Mohamed Riza[2], Alexis Jiaying Khng[4], Su-Ann Goh[2], Yi Wang[2], Han Boon Oh[1], Chi Hui Chin[6], Sing Cheer Kwek[7], Zhi Peng Zhang[8], Desmond Luan Seng Ong[6], Swee Tian Quek[9], Chuan Chien Tan[1¤], Hwee Lin Wee[2], Jingmei Li[3,4‡]*, Philip Tsau Choong Iau[1,5‡], Mikael Hartman[2,3,5‡]

1 Department of General Surgery, Ng Teng Fong General Hospital, Singapore, Singapore, 2 Saw Swee Hock School of Public Health, National University of Singapore and National University Health System, Singapore, Singapore, 3 Department of Surgery, Yong Loo Lin School of Medicine, National University of Singapore and National University Health System, Singapore, Singapore, 4 Laboratory of Women's Health and Genetics, Genome Institute of Singapore, Singapore, Singapore, 5 Department of Surgery, National University Hospital and National University Health System, Singapore, Singapore, 6 Jurong Polyclinic, National University Polyclinics and National University Health System, Singapore, Singapore, 7 Bukit Batok Polyclinic, National University Polyclinics and National University Health System, Singapore, Singapore, 8 Choa Chu Kang Polyclinic, National University Polyclinics and National University Health System, Singapore, Singapore, 9 Department of Diagnostic Imaging, National University Hospital and National University Health System, Singapore, Singapore

☺ These authors contributed equally to this work.
¤ Current address: SOG- CC Tan Breast, Thyroid & General Surgery, Gleneagles Medical Centre, Singapore, Singapore
‡ These authors also contributed equally to this work
* lijm1@gis.a-star.edu.sg

**Funding:** This study is funded by the JurongHealth Fund (reference number JHF-20-RE-003) and the Precision Health Research Singapore Clinical Implementation Pilot (PRECISE CIP) Fund. M.H. is

## Abstract

Routine mammography screening is currently the standard tool for finding cancers at an early stage, when treatment is most successful. Current breast screening programmes are one-size-fits-all which all women above a certain age threshold are encouraged to participate. However, breast cancer risk varies by individual. The BREAst screening Tailored for HEr (BREATHE) study aims to assess acceptability of a comprehensive risk-based personalised breast screening in Singapore. Advancing beyond the current age-based screening paradigm, BREATHE integrates both genetic and non-genetic breast cancer risk prediction tools to personalise screening recommendations. BREATHE is a cohort study targeting to recruit ~3,500 women. The first recruitment visit will include questionnaires and a buccal cheek swab. After receiving a tailored breast cancer risk report, participants will attend an in-person risk review, followed by a final session assessing the acceptability of our risk stratification programme. Risk prediction is based on: a) Gail model (non-genetic), b) mammographic density and recall, c) BOADICEA predictions (breast cancer predisposition genes), and d) breast cancer polygenic risk score. For national implementation of personalised risk-based breast screening, exploration of the acceptability within the target populace is critical, in addition to validated predication tools. To our knowledge, this is the first study to implement a comprehensive risk-based mammography screening programme in Asia. The

supported by the JurongHealth Fund, PRECISE CIP Fund, the Breast Cancer Prevention Programme under Saw Swee Hock School of Public Health Programme of Research Seed Funding (SSHSPH-Res-Prog-BCPP), Breast Cancer Screening Prevention Programme under Yong Loo Lin School of Medicine (NUHSRO/2020/121/BCSPP/LOA), National Medical Research Council Clinician Scientist Award (Senior Investigator Category, NMRC/CSA-SI/0015/2017), the National University Cancer Institute Singapore (NCIS) Centre Grant Programme (CGAug16M005), NCIS Ng Teng Fong General Hospital Collaborative grant (CGAug16C002) and Asian Breast Cancer Research Fund. J.Li is supported by the National Research Foundation Singapore (NRF-NRFF2017-02) and BMRC Central Research Fund (Applied Translational Research). The funders had no role in study design, data collection and analysis, decision to publish, or preparation of the manuscript.

**Competing interests:** The authors have declared that no competing interests exist.

BREATHE study will provide essential data for policy implementation which will transform the health system to deliver a better health and healthcare outcomes.

## Introduction

Population-based mammography endeavours to reduce mortality via early detection and prompt treatment [1–3]. Despite growing evidence of high heterogeneity of breast cancer risk within populations, breast cancer screening programmes commonly recommend starting mammography screening at age 40 or 50 [4]. Furthermore, mammographic screening itself has many limitations–over-diagnosis and overtreatment being prime among them [5]. While substantially increasing the number of cases of early-stage breast cancer detected, it only marginally reduces the rate at which women present with advanced cancer, as illustrated in the Cochrane reviews [6], Canadian National Breast Screening Study [7] and other studies [8,9]. This has generated international interest in a more risk-stratified approach to the current "one-size-fits all" population screening programmes [10–15].

BreastScreen Singapore, a nation-wide mammography screening programme in Singapore established in 2002 by the Health Promotion Board, invites women aged 50 to 69 to participate in the early detection of breast cancer. However, only 66% of the target group have reported to ever had a mammogram, and half of them do not adhere to the recommended biennial screening guideline (<30% of the target group were reported to attend mammogram every 2 years) [16]. Lukewarm responses to these initiatives have been attributed to a low perception of risk and misperceptions of risk factors and knowledge of breast cancer by women [16–21]. A number of studies have since proposed that risk-based screening may improve timeliness of screening. Furthermore, under the current age-based screening paradigm, approximately 30% of diagnosed breast cancer cases in Singapore are women of a younger age than the recommended screening age by the national guidelines [22]. The striking difference of ~10 years in the peak age for breast cancer in between Asian (40 to 50 years) and Western (60 to 70 years) prompts the need to reconsider screening approaches adapted from Western studies in Asia [23]. The design and adoption of risk-stratified approach to screening is needful for timely identification and treatment of these high-risk individuals.

Personalised screening enhances an age-based screening paradigm by tailoring screening recommendations to the individual's risk profile [24]. This reduces the rate of false positive results and over-diagnosis in lower risk individuals, thereby providing a more effective method to identify high risk individuals for intervention [25]. Currently, to identify high risk individuals, most screening programmes rely primarily on the evaluation of age, family history, clinical and lifestyle factors, and the testing of pathogenic variants in breast cancer predisposition genes [14,26].

Breast cancer is a multifactorial disease with both genetic and non-genetic risk factors. The Gail model (also known as the Breast Cancer Risk Assessment Tool) was first developed in 1989 for prediction using non-genetic risk factors in Whites, and has since been calibrated and validated for other ethnicities [27]. Furthermore, information from the first screen (i.e. mammographic density and false positive status) are indicators of elevated risk [28]. The validated Breast and Ovarian Analysis of Disease Incidence and Carrier Estimation Algorithm (BOADI-CEA) model is able to predict carriership of mutations in known breast cancer genes such as *BRCA1* and *BRCA2* [2,29–32].

Known pathogenic variants are rare. Due to cost issues, they are usually tested in only high risk individuals [33,34]. Common variants (i.e. single nucleotide polymorphism (SNP))

associated with breast cancer risk have been discovered through genome-wide association studies [35]. Individually, these SNPs have minimal effect on risk. However, Mavaddat *et al.* built a polygenic Risk Scores (PRS)–a tally of 313 SNPs–that emerged as a robust means to estimate an individual's risk of breast cancer [36,37]. The PRS (313 SNPs) is able to reliably predict breast cancer risk, with those in the top centile having a lifetime absolute risk of 32.6% [38]. This PRS has been validated in women of Asian descent [38]. Despite a growing body of evidence illustrating the utility of PRS in population screening programmes, policy implementation has been low [3]. While the Gail model [27], mammographic density [39] and breast cancer predisposition genes [40] have been incorporated into prior risk stratification studies, implementation of PRS is less common [3].

BREATHE is a landmark study aiming to contextualise a personalised, risk-based screening approach to the Asian population (specific aims are listed in Table 1). The present study endeavours to explore the acceptability and potential impact on changes in screening behaviour of the BREATHE risk-stratified screening programme as the first step towards policy implementation. With the cost-effectiveness of similar approaches validated [41], it is hoped that BREATHE will greatly enhance resource allocation and patient outcomes in the era of precision medicine.

## Materials and methods

### Study design and setting

BREATHE is a prospective multi-centre cohort to study a new modality of breast cancer screening in healthy Singaporean women aged between 35 and 59. Recruitment started in Oct 2021 and plans to recruit ~3,500 participants from two hospitals (Ng Teng Fong General Hospital and National University Hospital) and two polyclinics (Bukit Batok Polyclinic and Choa Chu Kang Polyclinic) over a period of two years. To achieve coverage of all age groups of interest, recruitment targets were allocated as such: 20% aged 35 to 39 years; 40% aged 40 to 49 years; and 40% aged 50 to 59 years. The proportion selected was based on the background population in the 2019 population report published on Singapore Department of Statistics. Participants will be on active follow-up for two years. In brief, enrolled participants will be asked (1) to provide a buccal swap for genotyping at study entry and (2) to answer various questionnaires and surveys at study entry and at the two follow-ups (at ~3 months and ~2 years after study entry) (Fig 1). All surveys are translated to the three major languages used in Singapore: Mandarin, Malay and Tamil.

### Identification of eligible participants

The study team will identify potential participants (1) through the response to our advertisements on BREATHE (posters, flyers [see S1 Appendix], and blog.nus.edu.sg/BREATHE) or

**Table 1. Specific aims of BREAst screening Tailored for HEr (BREATHE).**

| The primary aims of our study are to: |
|---|
| 1) Study the acceptability of risk stratification to aid women for decision making to attend regular screening |
| 2) Assess if risk-based screening will improve willingness to screen and recall rates |
| 3) Evaluate the cost-effectiveness of changing screening frequencies based on the risk-based BREATHE breast screening strategy over the current age-based paradigm. |

| The secondary aims of our study are: |
|---|
| 1) Assess the current level of breast cancer awareness, given the increasing breast cancer education in the recent decade |
| 2) Study the association between breast cancer perceptions (e.g. family history, age, having children) and compliance to regular breast cancer screening |
| 3) Study changes in breast cancer risk factors (e.g. number of children, menopausal status/age) |

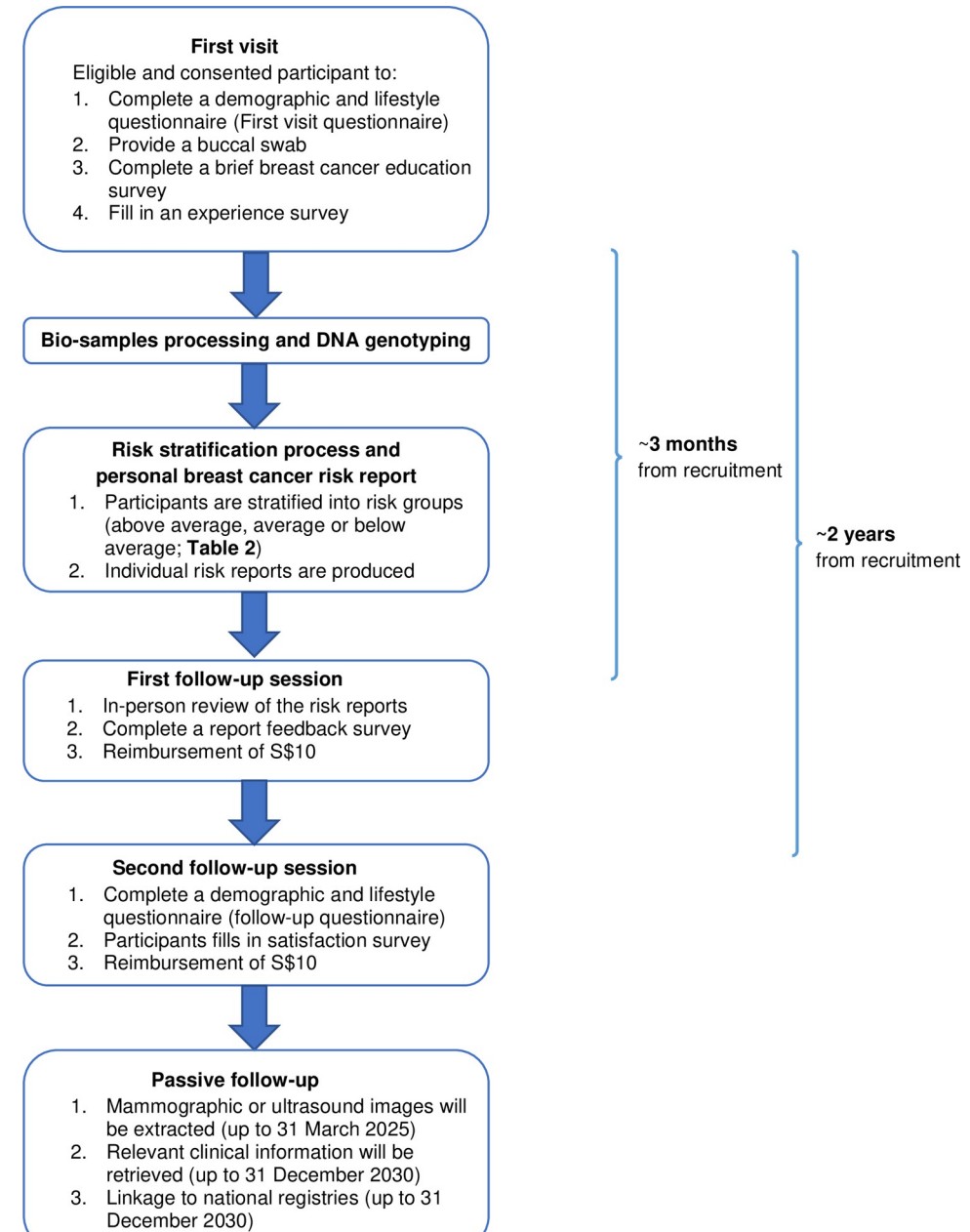

**Fig 1. Summary of the recruitment and follow-up process.** This study was approved by the National Healthcare Group Domain Specific Review. Board (reference no: 2020/01327). Written informed consent will be obtained from each participant.

(2) by approaching them at the participating institutions. The locations include diagnostic departments, women's clinics and waiting areas of the participating institutions. Responders to our advertisements can either call our hotline, email or fill up an online registration form (see S2 Appendix). They will be screened by study team members according to the eligibility criteria. Appointments will be scheduled for eligible participants to visit the participating hospitals or polyclinics for recruitment.

## Eligibility criteria

Participants must be Singapore Citizens or Permanent Residents, female, aged 35–59 years old. Women who have histologically confirmed diagnosis of any cancer, cognitive impairment which prevent the participant from giving voluntary consent, or are pregnant during recruitment will be excluded. Informed consent will be sought by trained study coordinators in the participant's language of choice (English, Chinese or Malay).

## First visit

After providing informed consent, participants will be asked to complete a demographic and lifestyle questionnaire and provide a buccal swab sample. A brief education session on breast cancer knowledge and the importance of regular and timely breast self-examination/screening will be self-administered by participants. Depending on their age, the participant will be advised to attend mammography (aged 40 years and above) breast screening. The session will end with a recruitment experience survey.

**First visit questionnaire.** Participants will fill in a structured questionnaire detailing various factors associated with the development of breast cancer and related conditions at baseline and over time. These include non-genetic risk factors (demographic, lifestyle, reproductive), past treatments and other environmental factors.

**Buccal swab, DNA extraction and genotyping.** Buccal swab (DNA Genotek, ORAcollect-DNA kit) samples will be de-identified and sent in batches (weekly) for deoxyribonucleic acid (DNA) extraction (QIAamp DNA Blood Midi Kit, Part No. 51185). Genotyping will be done (Illumina Global Screening Array [GSA-MD v3.0]) as per manufacturers' instructions.

DNA extracted from the bio-specimens will be stored in the freezer at -20 degrees Celsius for the duration of our research study. For participants who have agreed to the usage of their bio-specimens for future studies, DNA will be stored after study completion.

**Breast cancer education session.** A brief online education session will first assess the screening habits of the participants and their views about breast cancer (S3 Appendix). Various statements regarding breast cancer will be presented for participants to indicate their agreement. The correct answer and an accompanying explanation are given after every response submitted. The aim is to impart correct information about breast cancer and the importance of regular and timely breast self-examination/screening.

**Experience survey.** A short survey will be conducted to obtain feedback from the participants on their experience (including any discomfort) with the buccal swab and their initial views about risk-stratified breast cancer screening (S4 Appendix).

## Mammography screening

Participants may choose to attend screening within the next few months. The study coordinator will assist with setting up appointments for mammography screening with the participating institutions if required. If the participant (aged 40 and above) chooses to attend mammography screen, the study coordinator will seek consent to extract the mammogram image from the service provider (National Healthcare Group Diagnostics). Participants who had a recent mammogram (within one year prior to recruitment) done with National Healthcare Group Diagnostics can choose to provide consent for the study to extract the mammogram image.

## Risk stratification process and personal breast cancer risk report

Participants will be classified as above average, average or below average risk, based on (1) the Gail model; (2) information from the most recent mammography screening (mammography

**Table 2. Breast cancer risk reclassification criteria.**

Individuals who met **any one** of the following criteria will be considered above average risk:

- Predicted to be carriers of *BRCA1* or *BRCA2* by BOADICEA
- Extremely dense breast, which is ascertained according to the breast composition categories of the Breast Imaging-Reporting and Data System (5th edition)
- Positive recall status
- Gail model five-year absolute risk above 1.3%[a]
- Polygenic Risk Score (PRS) five-year absolute risk above 3%[b]

Individuals who met **all** of the following criteria will be considered below average risk:

- Age <50 years
- Gail model five-year absolute risk below 1.3%[a]
- PRS five-year absolute risk below 1.3% [a]

[a] The threshold of 1.3% is equivalent to the five-year absolute risk of developing breast cancer for an average Caucasian woman aged 50 years [42]. [b] The risk of an average *BRCA* carrier [43].

density and positive recall status); (3) BOADICEA; and (4) the PRS. Participants will first be considered average risk and reclassified as above average or below average based on the criteria in Table 2. A risk report will be produced and presented to the participant during the first follow-up session. All participants are recommended to follow current national guidelines (Table 3). In the BREATHE programme, women identified to be above average in breast cancer risk are referred to breast specialists at designated study sites, in addition to prevailing guidelines.

**Gail model (non-genetic risk factors).** The Gail Model requires the following breast cancer risk factors from the questionnaire from the first visit: age, age at menarche, age at first live birth, number of previous benign breast biopsies, presence of atypical hyperplasia on biopsy, family history of breast cancer (mother, sisters or daughters), and ethnicity [44,45]. Weights (logistic regression coefficients derived from the Gail model) and attributable risks of Asian-Americans will be used in the calculation of five-year absolute risk based on the Gail model ("Asian.AABCS", BCRA package in R) [45].

**Information from most recent mammography screening.** Mammographic density will be ascertained according to the breast composition categories of the Breast Imaging-Reporting and Data System (5th edition): almost entirely fatty, scattered areas of fibroglandular density, heterogeneously dense or extremely dense. A participant is considered recalled (i.e positive recall status) when she is asked to return for additional confirmatory examination or additional mammography views due to abnormal findings from initial screening.

**Breast and Ovarian Analysis of Disease Incidence and Carrier Estimation Algorithm (BOADICEA) predictions for breast cancer predisposition genes.** Carrier probabilities for breast cancer predisposition genes such as *BRCA1* and *BRCA2* will be predicted using the Breast and Ovarian Analysis of Disease Incidence and Carrier Estimation Algorithm (BOADICEA, web application v3, https://ccge.medschl.cam.ac.uk/boadicea/boadicea-web-application/, accessed Dec 28, 2021) [40]. Briefly, as described by Antoniou *et al* [40], the probability that

**Table 3. National guidelines for breast cancer screening in Singapore.**

| Age groups, years | National guidelines |
|---|---|
| 35 to 39 | No recommendation |
| 40 to 49 | Women are to attend yearly mammography screening, if recommended by their doctor. |
| 50 to 59 | Women are to attend mammography screening once every two years. |

an individual carries a mutation in *BRCA1/BRCA2* or other breast cancer genes based on family history can be computed using Bayes theorem.

**Breast cancer polygenic risk score.** PRS is estimated as the weighted sum of effect alleles in 313 SNPs found to be associated with breast cancer; using PLINK (version 3) with the "scoresum" option [46].

$$PRS = \beta_1 x_1 + \beta_2 x_2 + \cdots + \beta_k x_k + \cdots + \beta_{313} x_{313},$$

where $x_k$ is the risk allele (0, 1, 2) for SNP k, $\beta_k$ is the corresponding weight. The weights of the 313 SNPs for overall breast cancer risk were obtained from are of the overall breast cancer risk published by Mavaddat *et al* [47].

## First follow-up session

The first follow-up session occurs within three months of the recruitment date. This involves an in-person review of the risk reports and ends with a survey on their understanding of the risk report (S5 Appendix). Participants will be reimbursed S$10 for their time, inconvenience and transportation costs at the end of the first follow-up session.

## Second follow-up session

This is the final in-person follow-up conducted for all participants and occurs approximately two years from date of recruitment. The study coordinator will administer a questionnaire on non-genetic risk factors to capture any changes in participant characteristics since the first visit. The session ends with a satisfaction survey to understand the acceptability of our proposed risk stratification screening programme (S6 Appendix). Participants will be reimbursed S$10 for their time, inconvenience and transportation costs at the end of the final study visit.

## Passive follow-up

Mammogram images will be extracted if participants have undergone breast screening up to 31 March 2025. In addition, clinical information (e.g. radiology reports, medical conditions, medications and medical reports) related to this study will be retrieved from hospital/polyclinic medical records in accordance to the institutional guidelines, up to 31 December 2030. Clinical information may also be obtained through linkage to nation-wide health-related databases (Singapore Cancer Registry and the Registry of Births and Deaths), and may be done up to 31 December 2030.

## Planned statistical analysis

To gauge the acceptability of risk stratification (Primary Aim 1) and the current level of breast cancer awareness (Secondary Aim 1), descriptive statistics will be performed. Chi-square test for categorical variables and Kruskal-Wallis test for continuous variables will be used for testing differences among risk groups. Post-hoc analysis may be applied for pairwise comparisons. Information will be obtained from the recruitment experience survey (e.g. "I will recommend doing a breast cancer risk classification to others [options: strongly agree, agree, neither agree nor disagree, disagree, strongly disagree]"), report feedback survey (e.g. "I am confident that my personalized breast cancer risk profiles are reliable [options: strongly agree, agree, neither agree nor disagree, disagree, strongly disagree]"), and satisfaction survey (e.g. "What do you like about your experience in this study?") (Primary Aim 1); and from the breast cancer education questionnaire (e.g. "I am still young, therefore I do not need to screen for breast cancer [options: agree, disagree]") (Secondary Aim 1).

Logistic regression will be used to study the association between risk perceptions (i.e. risk categories and perceived risk) and follow-up events, which includes actual attendance of breast cancer screening and recall rates (Primary Aim 2 and Secondary Aim 2). Other modelling techniques will be employed dependent on event rates. Adjusted analysis may be done if variability in demographic variables are significant (e.g. conditional logistic models).

To understand potential short-term changes in breast cancer risk factors, paired analysis (e.g. paired-t-test, rank-sum test) between information from the first visit and follow-up will be performed (Secondary Aim 3).

Taking the healthcare system perspective, a cost-utility analysis will be conducted to compare BREATHE's recommendation with the prevailing breast cancer screening guidelines using a Markov model (Aim 3). Utility weights and various unit costs will be sourced from existing literature. Additional costs associated with breast cancer risk profiling, and changes in healthcare expenditure and health outcomes for different risk groups will be examined. The incremental cost-effectiveness ratio (incremental cost/incremental quality-adjusted life years) will be calculated to understand the cost-effectiveness of BREATH recommendations. The resulting model code and parameters will be made publicly available in an independent study.

## Discussion

Many previous works have evaluated the validity and discriminatory power of breast cancer risk calculators, alone or in combination [27,28,48]. In spite of the advances in breast cancer risk prediction, screening recommendations in practice have remained largely unchanged for the past few decades [23]. Several large-scale studies conducted in populations of European ancestry, such as KARMA—KARolinska MAmmography Project for Risk Prediction of Breast Cancer [49], PROCAS—Predicting the Risk of Cancer at Screening [50], WISDOM—Women Informed to Screen Depending On Measures of risk [43], are already underway to evaluate the feasibility of implementing risk stratification in breast screening programmes. However, prediction tools should be validated and calibrated to the target population [51]. To our knowledge, BREATHE is the first initiative to incorporate risk stratification approaches to enhance the efficacy of existing breast screening protocols in Asia.

Our study leverages on the existing national breast cancer screening programme (BreastScreen Singapore) [21]. Hence mammography service is consistent across all participants. The setup is scalable to include additional hospitals and polyclinics in the future. Singapore is geographically small and convenient for participants to visit breast clinics for recommendations to manage their breast cancer risk. While potential participants can visit multiple hospitals or polyclinics throughout our recruitment period, each individual's unique National Registration Identity Card number allows us to track them for follow-up. Loss to passive follow-up due to emigration is expected to be minimal for the duration of the study. The BREATHE risk classification is adapted from the established WISDOM Personalized Breast Cancer Screening Trial [43]. WISDOM uses a five-year absolute risk threshold of 6% (risk of an average BRCA carrier) for stratification [43]. However, it is known that the incidence of breast cancer among Asian women is lower [38,43]. Hence, the BREATHE study uses five-year absolute risk above 3% as a threshold (equivalent to women aged 50 years at the top risk percentile based on PRS in Singapore, S7 Appendix).

In this study, only one risk report is generated per participant based on information at recruitment. Over time, the participant's risk will change. An updated risk report is recommended if the programme were to be implemented in a nation-wide screening programme. As our risk estimates reflect a five-year absolute risk by the Gail model and PRS, a review of breast cancer risk should be done at a minimum frequency of five years.

Personal risk of breast cancer is a difficult concept to grasp. Breast cancer risk prediction tools are mainly designed for providers, hence the participant may have a limited understanding of the results [52]. We deliberated over the readiness of the Singapore's population to receive personalised disease risk results with various stakeholders (representatives of the hospitals and clinics, ethics representatives, public health experts, and researchers), and concluded that only the general risk classification (below-average, average and above-average risk) is to be conveyed to the participants as part of this research study.

Details on the risk classification categories are communicated and explained to the participants by healthcare professionals to ensure that the results are interpreted accurately and meaningfully. At recruitment, participants are briefed about the risk stratification tools used in this study. We understand that in this current information age, it is likely that there will be participants especially those in the above-average risk group will want details about their risk levels. Our main concern is in the participants' understanding of genetic risk, in particular the genetic risk component. As we want to create an inclusive risk-based screening program, we are not performing clinical genetic testing as part of this screening design. Unlike the better known and more expensive clinical genetic testing, PRS is a new method of breast cancer risk assessment that is not familiar to our study population. To avoid the situation where participants reject screening due to any misconception about risk (i.e. the idea that if I am genetically above-average risk there is nothing I can do to avoid getting breast cancer), we will only indicate the general risk level in the risk report. Based on the risk assessment by our ethics board, it is prudent that the follow-up of the above-average risk group and conveying of the details of the risk classification is done based on current clinical practice by clinicians and not part of the research study. Participants who are predicted carriers of *BRCA1/2* by BOADICEA will be recommended to see a breast specialist and referred to genetic counselling and subsequently genetic testing if appropriate.

We expect challenges in attaining our target recruitment of 3500 healthy women aged between 35 and 59. As this is a hospital led research study, publicity is limited to the hospitals and affiliated polyclinics. BREATHE may be extended to other sites (e.g. Alexandra Hospital and Jurong Medical Centre) to increase recruitment numbers. Participants are encouraged to refer family and friends to the program; they are given flyers with the sign-up link for enrolment. In addition, given the small geographic size of Singapore, redeployment of recruiters from sites with low numbers of potential participants to sites with higher numbers of potential participants is in our strategy to reach out to as many women as possible.

BREATHE has some limitations worth noting. Selection bias may arise due to systematic differences between baseline characteristics of responders and non-responders to BREATHE's advertisements. BREATHE participants may be more health conscious or are already attending breast screening. Such a bias may affect sample representativeness and generalizability of findings. However, the BREATHE study collects information on the study participants (e.g. profession, socio-economic status, highest education attained) and how they found out about the study. This information will allow us to assess the implementation of a risk-based screening approach in this population first, before rolling out the initiative on a larger scale. The BREATHE risk report is based on information available from each participant. For example, if the participant does not participate in or is ineligible for mammography screening, information from first screen will not be in the risk report (participant is assumed to be of average risk). When information is incomplete, breast cancer risk will be underestimated. Barriers to active follow-up two years later are expected. However, the study coordinators will be actively contacting the participants to remind them about the follow-up visit. In addition, the questionnaire is designed such that the participant does not need to be present in-person (conducted electronically or over a phone call).

## Conclusions

The aims of BREATHE are aligned with efforts to use personalized health for tailored interventions. For breast cancer screening, multiple studies have supported a risk-stratified approach over the current age-based paradigm due to potentially higher cost-effectiveness and reduced over-diagnosis [13,24,53,54]. If BREATHE is successful, women will gain a realistic understanding of their personal risk of breast cancer as well as strategies to reduce their risk, and fewer women will suffer from the anxiety of false positive mammograms and unnecessary biopsies. This work puts Singapore on the world map as a pioneer in integrating state-of-the-art breast cancer risk prediction tools, in particular, breast cancer PRS, in breast cancer screening. This study has real potential to transform breast cancer screening in Singapore.

BREATHE has assembled a multidisciplinary team to build on best practices and emerging data from other risk-based breast cancer screening studies elsewhere. Data-driven and patient-centric value-based care will benefit the healthcare system in many aspects. At the personal level: Women will gain a realistic understanding of their personal breast cancer risk and be empowered to make informed decisions together with their physicians on strategies to manage their risk. At the clinic: The comprehensive risk classification will aid physicians in the conversation on the need for further genetic testing as well as screening and risk reduction strategies. At the population-level: BREATHE generates real-world evidence on how to change the breast cancer screening paradigm to recognize the different needs of individuals. This includes assessment of the organizational readiness, effectiveness, efficiency, resources, costs and cost-effectiveness of implementing a risk-based breast cancer screening approach in Singapore. BREATHE puts Singapore on the world map as one of the pioneers in integrating state-of-the-art risk prediction tools in breast cancer screening, with a real potential to transform the health system to deliver better health and healthcare outcomes.

## Data availability

No datasets were generated or analysed as part of this submission. All relevant data from this study will be made available upon study completion. The data generated by this study is owned by the providing institutions (e.g. NTFGH, NUH, NUP). Data may be obtained with reasonable request to the main Principal Investigator Mikael Hartman (mikael_hartman@-nuhs.edu.sg). The data is not publicly available due to privacy or ethical restrictions. Legal agreements will need to be drawn up between data requesters and providers for access to the de-identified data. The proposed studies will need to be in compliance with Singapore's laws and regulations with regards to human biomedical research and clinical investigation including The Declaration of Helsinki, International Good Clinical Practice Guidelines, Good Clinical Practice guidelines by Singapore's Health Science Authority and the Ministry of Health.

## Supporting information

**S1 Appendix. Advertisement for BREATHE.**
(PDF)

**S2 Appendix. Online registration form for potential participants who are interested in BREATHE study.**
(PDF)

**S3 Appendix. Breast cancer education survey.**
(PDF)

**S4 Appendix. Experience survey.**
(PDF)

**S5 Appendix. Risk report feedback survey.**
(PDF)

**S6 Appendix. Satisfaction survey.**
(PDF)

**S7 Appendix. Obtaining appropriate threshold for the five-year absolute risk by polygenic risk score (PRS).**
(PDF)

## Acknowledgments

We want to thank our dedicated research and administrative staff—Ganga Devi D/O Chandrasegran, Hui Min Lau, Pooi Yee Wong, Hui Ling Tan, Kimiie Chia Wei Lin, Nabilah Binte Supiee, Nurfilya Binte Hamdil, Amanda Ong Tse Woon, Jing Jing Hong, Siew Li Tan, Evelyn Low Sok Peng, Marina Mohd, Noor Aisha Binte Mohamed Bahru Ali and Linus Chui for their contributions in the planning and preparation of BREATHE. We wish to acknowledge the contribution of Singapore Consortium of Cohort Studies-Multi-ethnic cohort (MEC) in providing information on women without breast cancer which is representative of the general population.

## Author Contributions

**Conceptualization:** Hwee Lin Wee, Jingmei Li, Philip Tsau Choong Iau, Mikael Hartman.

**Funding acquisition:** Jingmei Li, Mikael Hartman.

**Investigation:** Alexis Jiaying Khng, Jingmei Li, Philip Tsau Choong Iau, Mikael Hartman.

**Methodology:** Jenny Liu, Peh Joo Ho, Su-Ann Goh, Yi Wang, Hwee Lin Wee, Jingmei Li, Philip Tsau Choong Iau, Mikael Hartman.

**Project administration:** Jenny Liu, Yen Shing Yeoh, Ying Jia Chew, Nur Khaliesah Mohamed Riza.

**Resources:** Han Boon Oh, Chi Hui Chin, Sing Cheer Kwek, Zhi Peng Zhang, Desmond Luan Seng Ong, Swee Tian Quek, Chuan Chien Tan, Jingmei Li, Philip Tsau Choong Iau, Mikael Hartman.

**Supervision:** Han Boon Oh, Chi Hui Chin, Sing Cheer Kwek, Zhi Peng Zhang, Desmond Luan Seng Ong, Swee Tian Quek, Chuan Chien Tan, Jingmei Li, Philip Tsau Choong Iau, Mikael Hartman.

**Writing – original draft:** Jenny Liu, Peh Joo Ho, Tricia Hui Ling Tan, Jingmei Li.

**Writing – review & editing:** Jenny Liu, Peh Joo Ho, Tricia Hui Ling Tan, Yen Shing Yeoh, Ying Jia Chew, Nur Khaliesah Mohamed Riza, Alexis Jiaying Khng, Su-Ann Goh, Yi Wang, Han Boon Oh, Chi Hui Chin, Sing Cheer Kwek, Zhi Peng Zhang, Desmond Luan Seng Ong, Swee Tian Quek, Chuan Chien Tan, Hwee Lin Wee, Jingmei Li, Philip Tsau Choong Iau, Mikael Hartman.

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
