## [Decision Letter · Decision Letter 0]

23 Dec 2021

PONE-D-21-33800BREAst screening Tailored for HEr (BREATHE) - A study protocol on personalised risk-based breast cancer screening programmePLOS ONE

Dear Dr. Li,

Thank you for submitting your manuscript to PLOS ONE. After careful consideration, we feel that it has merit but does not fully meet PLOS ONE’s publication criteria as it currently stands. Therefore, we invite you to submit a revised version of the manuscript that addresses the points raised during the review process.

We look forward to receiving your revised manuscript.

Kind regards,

Yonglan Zheng, Ph.D.

Academic Editor

PLOS ONE

Journal Requirements:

Reviewers' comments:

Reviewer's Responses to Questions

**Comments to the Author**

1. Does the manuscript provide a valid rationale for the proposed study, with clearly identified and justified research questions?

Reviewer #1: Yes

Reviewer #2: Yes

2. Is the protocol technically sound and planned in a manner that will lead to a meaningful outcome and allow testing the stated hypotheses?

Reviewer #1: Partly

Reviewer #2: Yes

3. Is the methodology feasible and described in sufficient detail to allow the work to be replicable?

Reviewer #1: No

Reviewer #2: Yes

4. Have the authors described where all data underlying the findings will be made available when the study is complete?

Reviewer #1: No

Reviewer #2: Yes

5. Is the manuscript presented in an intelligible fashion and written in standard English?

Reviewer #1: Yes

Reviewer #2: Yes

6. Review Comments to the Author

You may also provide optional suggestions and comments to authors that they might find helpful in planning their study.

Reviewer #1: This study protocol reports on an important research endeavor. It starts by reiterating the clear case for risk stratification for breast cancer screening. It further notes that there is a lack of evidence on this topic for Asian populations.

The study has multiple aims. Its foundation is the recruitment of a cohort of 3,500 women aged 35 to 59 for whom a range of risk factors will be collected, including the latest in genetic markers. Based on these risk factors, the women in the cohort are being triaged into high, average, and low risk groups. They will be actively followed for 2 years, and passively followed via health care records until 2030. All women in the study will be asked to complete several questionnaires, and recommended to follow the current national guidelines (which are screening starting at age 40). Further, based on their assessed risk, those classified as high risk will be referred to a “breast specialist”.

The definition of high risk is cautious, in that any one of five criteria will be sufficient for a woman to be classified as high risk. However, there are two important questions about these criteria:

-- The BOADICEA model is being used to estimate the likelihood that the woman is a carrier of the BRCA1/2 mutations. However, recent versions of BOADICEA incorporate the likelihoods of 5 pathogenic variants plus the Mavaddat PRS plus a number of non-genetic risk factors, though this newer version of BOADICEA has not been validated in Asian populations. At the least, some discussion of the choice of the version of the BOADICEA model is needed.

-- The Gail model and PRS risk thresholds are expressed in terms of fixed absolute risk levels, independently of each woman’s age. This is a serious concern, as absolute breast cancer risk increases very rapidly with age. These thresholds will therefore over-estimate women at high risk at younger ages, and under-estimate those at higher ages.

The project has many and diverse aims, so there is a question of practical feasibility. The aims range from using questionnaires to assess women’s knowledge of and interests in risk stratified screening, to their actual responses (and implicitly those of their health care providers) to information about their assessed breast cancer risks, to a Markov model-based cost-utility analysis. Some indications of the scale of funding and scope of activities would therefore be helpful. For example, where will the utility weights and various unit costs for this latter analysis be sourced?

The paper notes the risks of sample selection bias in the recruitment of women for the cohort study, but makes no mention of the challenges of actually attaining the target sample size. Experience has shown that this can be a major problem. It is also not clear that the sample size will be sufficient to generate a large enough sample of high-risk women to provide adequate power to address the various aims of the study.

Regarding the PLOS questions, my reasoning is as follows:

2. The protocol will likely work, but the responses to some eventualities, like low sample recruitment rates or a seriously biased sample, are not described in sufficient detail. Similarly, the level of detail on the Markov cost-utility modeling is not adequate, though this could be remedied by undertaking to make the resulting model code and parameters fully public.

3. The methodology is likely feasible, but not described in sufficient detail to allow replicbility. Still, in such a large and complex study, this may not be a reasonable expectation. A very real challenge is the confidentiality of the cohort members’ health care records, so the data used for the final analysis will likely not be publicly available. This is not the fault of the authors, but rather a larger problem with this kind of research.

4. As just noted, it will likely be impossible for all the data to be made public, for reasons of patient confidentiality. However, it would be acceptable if the Singaporean authorities had a means whereby duly authorized independent researchers could access the full data under appropriate controls, as is possible in other countries.

Reviewer #2: The authors introduced the study protocol of personalized breast cancer screening programs. Various breast cancer risk prediction models were developed, but they have not been implemented in clinical practice. The study is essential for the implementation of personalized breast cancer screening.

1. According to the personalized risk evaluation, the participants will be classified into three risk levels. Can the participants know their reasons for risk classification? If participants have a high probability of familial breast cancer, they need genetic counseling and close follow-ups. Is there any rule for the potential familial breast cancer variant carriers?

2. The authors defined several primary and secondary aims in the study; however, the measures of the aims were not clear. How did the authors evaluate the acceptability of risk stratification (primary aim 1)? What is the measure for breast cancer awareness (secondary aim 1)?

7. PLOS authors have the option to publish the peer review history of their article (what does this mean?). If published, this will include your full peer review and any attached files.

Reviewer #1: **Yes: **Michael Wolfson

Reviewer #2: **Yes: **Isao Oze

---

## [Author Response · Author response to Decision Letter 0]

13 Jan 2022

Below are the responses to editor and reviewers. A same copy of Word document has been uploaded as "Response to Reviewers".

Journal Requirements:

Response: We have checked through our manuscript according to the style template provided and ensure our manuscript and file naming meet the style requirements. 

Response: We would like to update the ‘Financial Disclosure’ section to the following: “This study is funded by the JurongHealth Fund (reference number JHF-20-RE-003) and the Precision Health Research Singapore Clinical Implementation Pilot (PRECISE CIP) Fund. M.H. is supported by the JurongHealth Fund, PRECISE CIP Fund, the Breast Cancer Prevention Programme under Saw Swee Hock School of Public Health Programme of Research Seed Funding (SSHSPH-Res-Prog-BCPP), Breast Cancer Screening Prevention Programme under Yong Loo Lin School of Medicine (NUHSRO/2020/121/BCSPP/LOA), National Medical Research Council Clinician Scientist Award (Senior Investigator Category, NMRC/CSA-SI/0015/2017), the National University Cancer Institute Singapore Centre Grant Programme (CGAug16M005), and Asian Breast Cancer Research Fund. J.Li is supported by the National Research Foundation Singapore (NRF-NRFF2017-02) and BMRC Central Research Fund (Applied Translational Research). The funders had no role in study design, data collection and analysis, decision to publish, or preparation of the manuscript.” In addition, we have revised the grant information to ensure the information provided in the ‘Funding Information’ matches that in the ‘Financial Disclosure’ section.

Response: We have updated the data availability statement to state that there is no data generated for this submission and future data generated from this study will be available upon request. No datasets were generated or analysed as part of this submission. All relevant data from this study will be made available upon study completion. The data generated by this study is owned by the providing institutions (e.g. NTFGH, NUH, NUP). Data may be obtained with reasonable request to the main Principal Investigator Mikael Hartman (mikael_hartman@nuhs.edu.sg). The data is not publicly available due to privacy or ethical restrictions. Legal agreements will need to be drawn up between data requesters and providers for access to the de-identified data. The proposed studies will need to be in compliance with Singapore’s laws and regulations with regards to human biomedical research and clinical investigation including The Declaration of Helsinki, International Good Clinical Practice Guidelines, Good Clinical Practice guidelines by Singapore’s Health Science Authority and the Ministry of Health.

Response: We have included the information as a supplementary document (S7 Appendix).

Response: We have included the ethic statement in the methods section (page 8, line 155-157 in tracked changes version).

Response: All the figure files have been uploaded to PACE and passed the PLOS requirements. 

 

Review Comments to the Author:

Reviewer #1: This study protocol reports on an important research endeavor. It starts by reiterating the clear case for risk stratification for breast cancer screening. It further notes that there is a lack of evidence on this topic for Asian populations.

The study has multiple aims. Its foundation is the recruitment of a cohort of 3,500 women aged 35 to 59 for whom a range of risk factors will be collected, including the latest in genetic markers. Based on these risk factors, the women in the cohort are being triaged into high, average, and low risk groups. They will be actively followed for 2 years, and passively followed via health care records until 2030. All women in the study will be asked to complete several questionnaires, and recommended to follow the current national guidelines (which are screening starting at age 40). Further, based on their assessed risk, those classified as high risk will be referred to a “breast specialist”.

The definition of high risk is cautious, in that any one of five criteria will be sufficient for a woman to be classified as high risk. However, there are two important questions about these criteria:

-- The BOADICEA model is being used to estimate the likelihood that the woman is a carrier of the BRCA1/2 mutations. However, recent versions of BOADICEA incorporate the likelihoods of 5 pathogenic variants plus the Mavaddat PRS plus a number of non-genetic risk factors, though this newer version of BOADICEA has not been validated in Asian populations. At the least, some discussion of the choice of the version of the BOADICEA model is needed.

Response: BOADICEA Web Application (BWA) v3 (https://ccge.medschl.cam.ac.uk/boadicea/boadicea-web-application/, accessed Dec 28, 2021) will be used to compute BRCA mutation carrier probabilities. Prediction of other breast cancer predisposition genes is not available using the BWA. Gail model risk estimates and PRS are computed separately based on our understanding of Asian breast cancer risk based on Singaporean breast cancer incidence and mortality rates.

We have included this in (page 14, line 261-265 in tracked changes version): “Carrier probabilities for breast cancer predisposition genes such as BRCA1 and BRCA2 will be predicted using the Breast and Ovarian Analysis of Disease Incidence and Carrier Estimation Algorithm (BOADICEA, web application v3, https://ccge.medschl.cam.ac.uk/boadicea/boadicea-web-application/, accessed Dec 28, 2021) (40).”

-- The Gail model and PRS risk thresholds are expressed in terms of fixed absolute risk levels, independently of each woman’s age. This is a serious concern, as absolute breast cancer risk increases very rapidly with age. These thresholds will therefore over-estimate women at high risk at younger ages, and under-estimate those at higher ages.

Response: Our study looks at the short-term risk, i.e. the risk of breast cancer in the next five years from screening. As rightfully pointed out by the Reviewer, using the fixed threshold for absolute risk does not convey the risk of the woman throughout her lifetime. Re-assessment and conveying of changes in risk level should be done at five-years intervals. 

We have included additional information (page 19, line 369-376 in tracked changes version): “In this study, only one risk report is generated per participant based on information at recruitment. Over time, the participant’s risk will change. An updated risk report is recommended if the programme were to be implemented in a nation-wide screening programme. As our risk estimates reflect a five-year absolute risk by the Gail model and PRS, a review of breast cancer risk should be done at a minimum frequency of five years.”

The project has many and diverse aims, so there is a question of practical feasibility. The aims range from using questionnaires to assess women’s knowledge of and interests in risk stratified screening, to their actual responses (and implicitly those of their health care providers) to information about their assessed breast cancer risks, to a Markov model-based cost-utility analysis. Some indications of the scale of funding and scope of activities would therefore be helpful. For example, where will the utility weights and various unit costs for this latter analysis be sourced?

Response: Utility weights and various unit costs for the cost-effectiveness analysis will be sourced from existing literature.

We have included this in (page 17, line 331-332 in tracked changes version): “Utility weights and various unit costs will be sourced from existing literature. “

The paper notes the risks of sample selection bias in the recruitment of women for the cohort study, but makes no mention of the challenges of actually attaining the target sample size. Experience has shown that this can be a major problem. It is also not clear that the sample size will be sufficient to generate a large enough sample of high-risk women to provide adequate power to address the various aims of the study.

Response: BREATHE will be expanded to cover two more sites with additional funding. Staggered opening of sites was done to ensure sufficient effort is provided to smoothen out any logistics issues. We have recruited 286 women after two months from the opening of the first recruitment site. Additional funding has been secured to expand BREATHE to two additional sites. Within the first 100 participants, 20% were identified as at above-average risk. This is higher than expected; in existing data sources we are expecting ~12% of women to be at above-average risk. Selection bias is evident and the recruiters have been briefed to not only target the subpopulations of women who are at the mammogram clinics but also at other waiting areas (e.g. for other purposes than health screening) of the polyclinics. 

We have included this in (page 20-21, line 407-415 in tracked changes version): “We expect challenges in attaining our target recruitment of 3500 healthy women aged between 35 and 59. As this is a hospital led research study, publicity is limited to the hospitals and affiliated polyclinics. BREATHE may be extended to other sites (e.g. Alexandra Hospital and Jurong Medical Centre) to increase recruitment numbers. Participants are encouraged to refer family and friends to the program; they are given flyers with the sign-up link for enrolment. In addition, given the small geographic size of Singapore, redeployment of recruiters from sites with low numbers of potential participants to sites with higher numbers of potential participants is in our strategy to reach out to as many women as possible.”

Regarding the PLOS questions, my reasoning is as follows:

2. The protocol will likely work, but the responses to some eventualities, like low sample recruitment rates or a seriously biased sample, are not described in sufficient detail. Similarly, the level of detail on the Markov cost-utility modeling is not adequate, though this could be remedied by undertaking to make the resulting model code and parameters fully public.

Response: 

We have included this in (page 17, line 336-337 in tracked changes version): “The resulting model code and parameters will be made publicly available in an independent study.”

3. The methodology is likely feasible, but not described in sufficient detail to allow replicbility. Still, in such a large and complex study, this may not be a reasonable expectation. A very real challenge is the confidentiality of the cohort members’ health care records, so the data used for the final analysis will likely not be publicly available. This is not the fault of the authors, but rather a larger problem with this kind of research.

Response: "The data generated by this study is owned by the providing institutions (e.g. NTFGH, NUH, NUP). Data may be obtained with reasonable request to the main Principal Investigator Mikael Hartman (mikael_hartman@nuhs.edu.sg). The data is not publicly available due to privacy or ethical restrictions. Legal agreements will need to be drawn up between data requesters and providers for access to the de-identified data. The proposed studies will need to be in compliance with Singapore’s laws and regulations with regards to human biomedical research and clinical investigation including The Declaration of Helsinki, International Good Clinical Practice Guidelines, Good Clinical Practice guidelines by Singapore’s Health Science Authority and the Ministry of Health."

We have included the above text in the “Data availability” section.

4. As just noted, it will likely be impossible for all the data to be made public, for reasons of patient confidentiality. However, it would be acceptable if the Singaporean authorities had a means whereby duly authorized independent researchers could access the full data under appropriate controls, as is possible in other countries.

Response: The data generated by this study is owned by the providing institutions (NTFGH, NUH, NUP). Legal agreements may be drawn up between data requesters and providers for access to the de-identified data. For example, in this current study, the collaborator GIS has a legal agreement with the providers to use the data generated from this study for analysis. The proposed study will need to be in compliance with Singapore’s laws and regulations with regards to human biomedical research and clinical investigation including The Declaration of Helsinki, International Good Clinical Practice Guidelines, Good Clinical Practice guidelines by Singapore’s Health Science Authority and the Ministry of Health. 

 

Reviewer #2: The authors introduced the study protocol of personalized breast cancer screening programs. Various breast cancer risk prediction models were developed, but they have not been implemented in clinical practice. The study is essential for the implementation of personalized breast cancer screening.

1. According to the personalized risk evaluation, the participants will be classified into three risk levels. Can the participants know their reasons for risk classification? If participants have a high probability of familial breast cancer, they need genetic counseling and close follow-ups. Is there any rule for the potential familial breast cancer variant carriers?

Response: Participants are briefed about the process of risk classification, which includes the tools used (i.e. the Gail model, Polygenic Risk Scores, BOADICEA, mammogram density and recall status). Participants in the below-average and average risk category will know they are below-average or average risk based on all available information they provided. Participants in the above-average risk category will know their reasons for being at elevated risk if they attend the recommended clinic visit with a breast specialist. Based on risk assessment by our ethics board, it is prudent that the follow-up of the above-average risk group and conveying of why the participant is this risk level is done based on clinical practice by clinicians and not part of the research study. The research study will have access to the medical records to follow up on the recommendations by the breast specialist. Only participants who chose to attend the clinic incurring out-of-pocket expenses will know their risk, the reason is to simulate a nationwide screening scenario. 

We have included this in (page 19-20, line 378-405 in tracked changes version): “Personal risk of breast cancer is a difficult concept to grasp. Breast cancer risk prediction tools are mainly designed for providers, hence the participant may have a limited understanding of the results (52). We deliberated over the readiness of the Singapore’s population to receive personalised disease risk results with various stakeholders (representatives of the hospitals and clinics, ethics representatives, public health experts, and researchers), and concluded that only the general risk classification (below-average, average and above-average risk) is to be conveyed to the participants as part of this research study. Details on the risk classification categories are communicated and explained to the participants by healthcare professionals to ensure that the results are interpreted accurately and meaningfully. At recruitment, participants are briefed about the risk stratification tools used in this study. We understand that in this current information age, it is likely that there will be participants especially those in the above-average risk group will want details about their risk levels. Our main concern is in the participants’ understanding of genetic risk, in particular the genetic risk component. As we want to create an inclusive risk-based screening program, we are not performing clinical genetic testing as part of this screening design. Unlike the better known and more expensive clinical genetic testing, PRS is a new method of breast cancer risk assessment that is not familiar to our study population. To avoid the situation where participants reject screening due to any misconception about risk (i.e. the idea that if I am genetically above-average risk there is nothing I can do to avoid getting breast cancer), we will only indicate the general risk level in the risk report. Based on the risk assessment by our ethics board, it is prudent that the follow-up of the above-average risk group and conveying of the details of the risk classification is done based on current clinical practice by clinicians and not part of the research study. Participants who are predicted carriers of BRCA1/2 by BOADICEA will be recommended to see a breast specialist and referred to genetic counselling and subsequently genetic testing if appropriate.”

2. The authors defined several primary and secondary aims in the study; however, the measures of the aims were not clear. How did the authors evaluate the acceptability of risk stratification (primary aim 1)? What is the measure for breast cancer awareness (secondary aim 1)?

Response: We are using surveys tailored to Singapore’s population. The questionnaires are not validated, however, they are designed with reference to various publicity materials on breast cancer available in Singapore. We have included these surveys as supplementary materials (S3 Appendix – S6 Appendix). 

The acceptability of risk stratification is measured by our recruitment experience survey (done at the end of recruitment, S4 Appendix), report feedback survey (done after the return of risk report ~3 months after recruitment, S5 Appendix), and the satisfaction survey (done at the end of the study participation, ~2 years after recruitment, S6 Appendix). The three surveys ask questions about their physical and emotional experience 1) during recruitment, 2) while waiting for the risk report, 3) their understanding of the risk report, and 4) their thoughts about risk stratification in terms of inclusion of the PRS. Some questions include: “Since receiving my personalized breast cancer risk profile I feel_________”, “I will follow the recommended breast cancer screening frequency (options: strongly agree, agree, neither agree nor disagree, disagree, strongly disagree)”. “I am confident that my personalized breast cancer risk profiles are reliable (options: strongly agree, agree, neither agree nor disagree, disagree, strongly disagree)”.

We will be using a descriptive method to understand the current breast cancer awareness in Singapore. At recruitment, we will administer a breast cancer education questionnaire (S3 Appendix) including questions “I am still young, therefore I do not need to screen for breast cancer [options: agree, disagree]” and “If breast cancer is detected early, chances of surviving is high [options: agree, disagree]”.

We have included this in (page 16, line 308-316 in tracked changes version): “Information will be obtained from the recruitment experience survey (e.g. “I will recommend doing a breast cancer risk classification to others [options: strongly agree, agree, neither agree nor disagree, disagree, strongly disagree]”), report feedback survey (e.g. “I am confident that my personalized breast cancer risk profiles are reliable [options: strongly agree, agree, neither agree nor disagree, disagree, strongly disagree]”), and satisfaction survey (e.g. “What do you like about your experience in this study?”) (Primary Aim 1); and from the breast cancer education questionnaire (e.g. “I am still young, therefore I do not need to screen for breast cancer [options: agree, disagree]”) (Secondary Aim 1).”

---

## [Decision Letter · Decision Letter 1]

11 Mar 2022

BREAst screening Tailored for HEr (BREATHE) - A study protocol on personalised risk-based breast cancer screening programme

PONE-D-21-33800R1

Dear Dr. Li,

We’re pleased to inform you that your manuscript has been judged scientifically suitable for publication and will be formally accepted for publication once it meets all outstanding technical requirements.

Kind regards,

Yonglan Zheng, Ph.D.

Academic Editor

PLOS ONE

Reviewers' comments:

Reviewer's Responses to Questions

**Comments to the Author**

1. Does the manuscript provide a valid rationale for the proposed study, with clearly identified and justified research questions?

Reviewer #2: Yes

2. Is the protocol technically sound and planned in a manner that will lead to a meaningful outcome and allow testing the stated hypotheses?

Reviewer #2: Yes

3. Is the methodology feasible and described in sufficient detail to allow the work to be replicable?

Reviewer #2: Yes

4. Have the authors described where all data underlying the findings will be made available when the study is complete?

Reviewer #2: Yes

5. Is the manuscript presented in an intelligible fashion and written in standard English?

Reviewer #2: Yes

6. Review Comments to the Author

You may also provide optional suggestions and comments to authors that they might find helpful in planning their study.

Reviewer #2: The authors clearly answered all comments from the reviewer 2.

I have no more comments to the article.

7. PLOS authors have the option to publish the peer review history of their article (what does this mean?). If published, this will include your full peer review and any attached files.

Reviewer #2: No

---

## [Editor Report · Acceptance letter]

23 Mar 2022

PONE-D-21-33800R1 

BREAst screening Tailored for HEr (BREATHE) - A study protocol on personalised risk-based breast cancer screening programme 

Dear Dr. Li:

I'm pleased to inform you that your manuscript has been deemed suitable for publication in PLOS ONE. Congratulations! Your manuscript is now with our production department. 

Kind regards, 

on behalf of

Dr. Yonglan Zheng 

Academic Editor

PLOS ONE